# Hybrid Retrieval-Generation Reinforced Agent for Medical Image Report Generation

**Christy Y. Li**[*]
Duke University
yl558@duke.edu

**Xiaodan Liang**[†]
Carnegie Mellon University
xiaodan1@cs.cmu.edu

**Zhiting Hu**
Carnegie Mellon University
zhitingh@cs.cmu.edu

**Eric P. Xing**
Petuum, Inc
epxing@cs.cmu.edu

## Abstract

Generating long and coherent reports to describe medical images poses challenges to bridging visual patterns with informative human linguistic descriptions. We propose a novel Hybrid Retrieval-Generation Reinforced Agent (HRGR-Agent) which reconciles traditional retrieval-based approaches populated with human prior knowledge, with modern learning-based approaches to achieve structured, robust, and diverse report generation. HRGR-Agent employs a hierarchical decision-making procedure. For each sentence, a high-level *retrieval policy module* chooses to either retrieve a template sentence from an off-the-shelf template database, or invoke a low-level *generation module* to generate a new sentence. HRGR-Agent is updated via reinforcement learning, guided by sentence-level and word-level rewards. Experiments show that our approach achieves the state-of-the-art results on two medical report datasets, generating well-balanced structured sentences with robust coverage of heterogeneous medical report contents. In addition, our model achieves the highest detection precision of medical abnormality terminologies, and improved human evaluation performance.

## 1  Introduction

Beyond the traditional visual captioning task [41, 28, 43, 40, 18] that produces one single sentence, generating long and topic-coherent stories or reports to describe visual contents (images or videos) has recently attracted increasing research interests [19, 35, 22], posed as a more challenging and realistic goal towards bridging visual patterns with human linguistic descriptions. Particularly, report generation has several challenges to be resolved: 1) The generated report is a long narrative consisting of multiple sentences or paragraphs, which must have a plausible logic and consistent topics; 2) There is a presumed content coverage and specific terminology/phrases, depending on the task at hand. For example, a sports game report should describe competing teams, wining points, and outstanding players [38]. 3) The content ordering is very crucial. For example, a sports game report usually talks about the competition results before describing teams and players in detail.

As one of the most representative and practical report generation task, the desired medical image report generation must satisfy more critical protocols and ensure the correctness of medical term usage. As shown in Figure 1, a medical report consists of a *findings* section describing medical observations in details of both normal and abnormal features, an *impression* or *conclusion* sentence

---

[*]This work was conducted when Christy Y. Li was at Petuum, Inc.
[†]Corresponding author.

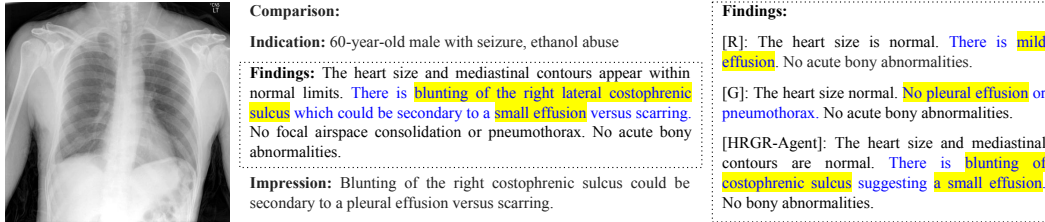

**Comparison:**

**Indication:** 60-year-old male with seizure, ethanol abuse

**Findings:** The heart size and mediastinal contours appear within normal limits. There is blunting of the right lateral costophrenic sulcus which could be secondary to a small effusion versus scarring. No focal airspace consolidation or pneumothorax. No acute bony abnormalities.

**Impression:** Blunting of the right costophrenic sulcus could be secondary to a pleural effusion versus scarring.

**Findings:**

[R]: The heart size is normal. There is mild effusion. No acute bony abnormalities.

[G]: The heart size normal. No pleural effusion or pneumothorax. No acute bony abnormalities.

[HRGR-Agent]: The heart size and mediastinal contours are normal. There is blunting of costophrenic sulcus suggesting a small effusion. No bony abnormalities.

Figure 1: An example of medical image report generation. The middle column is a report written by radiologists for the chest x-ray image on the left column. The right column contains three reports generated by a retrieval-based system (R), a generation-based model (G) and our proposed model (HRGR-Agent) respectively. The retrieval-based model correctly detects effusion while the generative model fails to. Our HRGR-Agent detects effusion and also describes supporting evidence.

indicating the most prominent medical observation or conclusion, and *comparison* and *indication* sections that list patient's peripheral information. Among these sections, the *findings* section posed as the most important component, ought to cover contents of various aspects such as heart size, lung opacity, bone structure; any abnormality appearing at lungs, aortic and hilum; and potential diseases such as effusion, pneumothorax and consolidation. And, in terms of content ordering, the narrative of *findings* section usually follows a presumptive order, e.g. heart size, mediastinum contour followed by lung opacity, remarkable abnormalities followed by mild or potential abnormalities.

State-of-the-art caption generation models [41, 9, 43, 34] tend to perform poorly on medical report generation with specific content requirements due to several reasons. First, medical reports are usually dominated by normal findings, that is, a small portion of majority sentences usually forms a template database. For these normal cases, a retrieval-based system (e.g. directly perform classification among a list of majority sentences given image features) can perform surprisingly well due to the low variance of language. For instance, in Figure 1, a retrieval-based system correctly detects effusion from a chest x-ray image, while a generative model that generates word-by-word given image features, fails to detect effusion. On the other hand, abnormal findings which are relatively rare and remarkably diverse, however, are of higher importance. Current text generation approaches [16] often fail to capture the diversity of such small portion of descriptions, and pure generation pipelines are biased towards generating plausible sentences that look natural by the language model but poor at finding visual groundings [17]. On the contrary, a desirable medical report usually has to not only describe normal and abnormal findings, but also support itself by visual evidences such as location and attributes of the detected findings appearing in the image.

Inspired by the fact that radiologists often follow templates for writing reports and modify them accordingly for each individual case [5, 12, 10], we propose a Hybrid Retrieval-Generation Reinforced Agent (HRGR-Agent) which is the first attempt to incorporate human prior knowledge with learning-based generation for medical reports. HRGR-Agent employs a *retrieval policy module* to decide between automatically generating sentences by a *generation module* and retrieving specific sentences from the template database, and then sequentially generates multiple sentences via a hierarchical decision-making. The template database is built based on human prior knowledge collected from available medical reports. To enable effective and robust report generation, we jointly train the *retrieval policy module* and *generation module* via reinforcement learning (RL) [30] guided by sentence-level and word-level rewards, respectively. Figure 1 shows an example generated report by our HRGR-Agent which correctly describes "a small effusion" from the chest x-ray image, and successfully supports its finding by providing the appearance ("blunting") and location ("costophrenic sulcus") of the evidence.

Our main contribution is to bridge rule-based (retrieval) and learning-based generation via reinforcement learning, which can achieve plausible, correct and diverse medical report generation. Moreover, our HRGR-Agenet has several technical merits compared to existing retrieval-generation-based models: 1) our retrieval and generation modules are updated and benefit from each other via policy learning; 2) the retrieval actions are regarded as a part of the generation whose selection of templates directly influences the final generated result. 3) the generation module is encouraged to learn diverse and complicated sentences while the retrieval policy module learns template-like sentences, driven by distinct word-level and sentence-level rewards, respectively. Other work such as [24] still enforces the generative model to predict template-like sentences.

We conduct extensive experiments on two medical image report dataset [8]. Our HRGR-Agent achieves the state-of-the-art performance on both datasets under three kinds of evaluation metrics: automatic metrics such as CIDEr [33], BLEU [25] and ROUGE [20], human evaluation, and detection precision of medical terminologies. Experiments show that the generated sentences by HRGR-Agent shares a descent balance between concise template sentences, and complicated and diverse sentences.

## 2   Related Work

**Visual Captioning and Report Generation.** Visual captioning aims at generating a descriptive sentence for images or videos. State-of-the-art approaches use CNN-RNN architectures and attention mechanisms [27, 41, 43, 28]. The generated sequence is usually short, describing only the dominating visual event, and is primarily rewarded by language fluency in practice. Generating reports that are informative and have multiple sentences [38, 16] poses higher requirements on content selection, relation generation, and content ordering. The task differs from image captioning [43, 23] and sentence generation [14, 6] where usually single or few sentences are required, or summarization [2, 44] where summaries tend to be more diverse without clear template sentences. State-of-the-art methods on report generation [16] are still remarkably cloning expert behaviour, and incapable of diversifying language and depicting rare but prominent findings. Our approach prevents from mimicking teacher behaviour by sparing the burden of automatic generative model with a template selection and retrieval mechanism, which by design promotes language diversity and better content selection.

**Template Based Sequence Generation.** Some of the recent approaches bridged generative language approaches and traditional template-based methods. However, state-of-the-art approaches either treat a retrieval mechanism as latent guidance [44], the impact of which to text generation is limited, or still encourage the generation network to mimic template-like sequences [24]. Our method is close to previous copy mechanism work such as pointer-generator [2], however, we are different in that: 1) our retrieval module aims to retrieve from an external common template base, which is particularly effective to the task, as opposed to copying from a specific source article; 2) we formulate the retrieval-generation choices as discrete actions (as opposed to soft weights as in previous work) and learn with hierarchical reinforcement learning for optimizing both short- and long-term goals.

**Reinforcement Learning for Sequence Generation.** Recently, reinforcement learning (RL) has been receiving increasing popularity in sequence generation [27, 3, 13] such as visual captioning [21, 28, 18], text summarization [26], and machine translation [39]. Traditional methods use cross entropy loss which is prone to exposure bias [27, 31] and do not necessarily optimize evaluation metrics such as CIDEr [33], ROUGE [20], BLEU [25] and METEOR [4]. In contrast, reinforcement learning can directly use the evaluation metrics as reward and update model parameters via policy gradient. There has been some recent efforts [42] devoted in applying hierarchical reinforcement learning (HRL) [7] where sequence generation is broken down into several sub-tasks each of which targets at a chunk of words. However, HRL for long report generation is still under-explored.

## 3   Approach

Medical image report generation aims at generating a report consisting of a sequence of sentences $\mathbf{Y} = (\mathbf{y}_1, \mathbf{y}_2, \ldots, \mathbf{y}_M)$ given a set of medical images $\mathbf{I} = \{I_j\}_{j=1}^K$ of a patient case. Each sentence comprises a sequence of words $\mathbf{y}_i = (y_{i,1}, y_{i,2}, \ldots, y_{i,N}), y_{i,j} \in \mathbb{V}$ where $i$ is the index of sentences, $j$ the index of words, and $\mathbb{V}$ the vocabulary of all output tokens. In order to generate long and topic-coherent reports, we formulate the decoding process in a hierarchical framework that first produces a sequence of hidden sentence topics, and then predicts words of each sentence conditioning on each topic.

It is observed that doctors writing a report tend to follow certain patterns and reuse templates, while adjusting statements for each individual case when necessary. To mimic the procedure, we propose to combine retrieval and generation for automatic report generation. In particular, we first compile an off-the-shelf template database $\mathbb{T}$ that consists of a set of sentences that occur frequently in the training corpus. Such sentences typically describe general observations, and are often inserted into medical reports, e.g., "the heart size is normal" and "there is no pleural effusion or pneumothorax". (Table 1 provides more examples.)

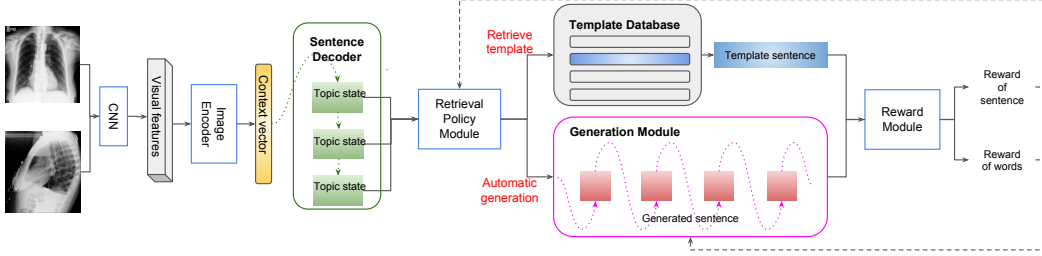

Figure 2: Hybrid Retrieval-Generation Reinforced Agent. Visual features are encoded by a CNN and image encoder, and fed to a sentence decoder to recurrently generate hidden topic states. A retrieval policy module decides for each topic state to either automatic generate a sentence, or retrieve a specific template from a template database. Dashed black lines indicate hierarchical policy learning.

As described in Figure 2, a set of images for each sample is first fed into a CNN to extract visual features which is then transformed into a context vector by an *image encoder*. Then a *sentence decoder* recurrently generates a sequence of hidden states $\mathbf{q} = (\mathbf{q}_1, \mathbf{q}_2, \ldots, \mathbf{q}_M)$ which represent sentence topics. Given each topic state $\mathbf{q}_i$, a *retrieval policy module* decides to either automatically generate a new sentence by invoking a *generation module*, or retrieve an existing template from the template database. Both the retrieval policy module (that determines between automatic generation or template retrieval) and the generation module (that generates words) are making discrete decisions and be updated via the REINFORCE algorithm [37, 30]. We devise sentence-level and word-level rewards accordingly for the two modules, respectively.

## 3.1 Hybrid Retrieval-Generation Reinforced Agent

**Image Encoder.** Given a set of images $\{I_j\}_{j=1}^{K}$, we first extract their features $\{\mathbf{v}_j\}_{j=1}^{K}$ with a pretrained CNN, and then average $\{\mathbf{v}_j\}_{j=1}^{K}$ to obtain $\mathbf{v}$. The image encoder converts $\mathbf{v}$ into a context vector $\mathbf{h}^v \in \mathbb{R}^D$ which is used as the visual input for all subsequent modules. Specifically, the image encoder is parameterized as a fully-connected layer, and the visual features are extracted from the last convolution layer of a DenseNet [15] or VGG-19 [29].

**Sentence Decoder.** Sentence decoder comprises stacked RNN layers which generates a sequence of topic states $\mathbf{q}$. We equip the stacked RNNs with attention mechanism to enhance text generation, inspired by [32, 41, 23]. Each stacked RNN first generates an attentive context vector $\mathbf{c}_i^s$, where $i$ indicates time steps, given the image context vector $\mathbf{h}^v$ and previous hidden state $\mathbf{h}_{i-1}^s$. It then generates a hidden state $\mathbf{h}_i^s$ based on $\mathbf{c}_i^s$ and $\mathbf{h}_{i-1}^s$. The generated hidden state $\mathbf{h}_i^s$ is further projected into a topic space as $\mathbf{q}_i$ and a *stop control* probability $z_i \in [0, 1]$ through non-linear functions respectively. Formally, the sentence decoder can be written as:

$$\mathbf{c}_i^s = F_{\text{attn}}^s(\mathbf{h}^v, \mathbf{h}_{i-1}^s) \tag{1}$$

$$\mathbf{h}_i^s = F_{\text{RNN}}^s(\mathbf{c}_i^s, \mathbf{h}_{i-1}^s) \tag{2}$$

$$\mathbf{q}_i = \sigma(\mathbf{W}_q \mathbf{h}_i^s + \mathbf{b}_q) \tag{3}$$

$$z_i = \text{Sigmoid}(\mathbf{W}_z \mathbf{h}_i^s + \mathbf{b}_z), \tag{4}$$

where $F_{\text{attn}}^s$ denotes a function of the attention mechanism [28], $F_{\text{RNN}}^s$ denotes the non-linear functions of Stacked RNN, $\mathbf{W}_q$ and $\mathbf{b}_q$ are parameters which project hidden states into the topic space while $\mathbf{W}_z$ and $\mathbf{b}_z$ are parameters for stop control, and $\sigma$ is a non-linear activation function. The stop control probability $z_i$ greater than or equal to a predefined threshold (e.g. 0.5) indicates stopping generating topic states, and thus the hierarchical report generation process.

**Retrieval Policy Module.** Given each topic state $\mathbf{q}_i$, the retrieval policy module takes two steps. First, it predicts a probability distribution $\mathbf{u}_i \in R^{1+|\mathbb{T}|}$ over actions of generating a new sentence and retrieving from $|\mathbb{T}|$ candidate template sentences. Based on the prediction of the first step, it triggers different actions. If automatic generation obtains the highest probability, the generation module is activated to generate a sequence of words conditioned on current topic state (the second row on the right side of Figure 2). If a template in $\mathbb{T}$ obtains the highest probability, it is retrieved from the off-the-shelf template database and serves as the generation result of current sentence topic (the first row on the right side of Figure 2). We reserve 0 index to indicate the probability of selecting automatic generation and positive integers in $\{1, |\mathbb{T}|\}$ to index the probability of selecting templates

in $\mathbb{T}$. The first step is parameterized as a fully-connected layer with Softmax activation:

$$\mathbf{u}_i = \text{Softmax}(\mathbf{W}_u\mathbf{q}_i + \mathbf{b}_u) \tag{5}$$
$$m_i = \text{argmax}(\mathbf{u}_i), \tag{6}$$

where $\mathbf{W}_u$ and $\mathbf{b}_u$ are network parameters, and the resulting $m_i$ is the index of highest probability in $\mathbf{u}_i$.

**Generation Module.** Generation module generates a sequence of words conditioned on current topic state $\mathbf{q}_i$ and image context vector $\mathbf{h}^v$ for each sentence. It comprises RNNs which take environment parameters and previous hidden state $\mathbf{h}^g_{i,t-1}$ as input, and generate a new hidden state $\mathbf{h}^g_{i,t}$ which is further transformed to a probability distribution $\mathbf{a}_{i,t}$ over all words in $\mathbb{V}$, where $t$ indicates $t$-th word. We define environment parameters as a concatenation of current topic state $\mathbf{q}_i$, context vector $\mathbf{c}^g_{i,t}$ encoded by following the same attention paradigm in sentence decoder, and embedding of previous word $\mathbf{e}_{i,t-1}$. The procedure of generating each word is written as follows, which is an attentional decoding step:

$$\mathbf{c}^g_{i,t} = F^g_{\text{attn}}(\mathbf{h}^v, [\mathbf{e}_{i,t-1}; \mathbf{q}_i], \mathbf{h}^g_{i,t-1}) \tag{7}$$
$$\mathbf{h}^g_{i,t} = F^g_{\text{RNN}}([\mathbf{c}^g_{i,t}; \mathbf{e}_{i,t-1}; \mathbf{q}_i], \mathbf{h}^g_{i,t-1}) \tag{8}$$
$$\mathbf{a}_t = \text{Softmax}(\mathbf{W}_y\mathbf{h}^g_{i,t} + \mathbf{b}_y) \tag{9}$$
$$y_t = \text{argmax}(\mathbf{a}_t) \tag{10}$$
$$\mathbf{e}_{i,t} = \mathbf{W}_e\mathbb{O}(y_{i,t}), \tag{11}$$

where $F^g_{\text{attn}}$ denotes the attention mechanism of generation module, $F^g_{\text{RNN}}$ denotes non-linear functions of RNNs, $\mathbf{W}_y$ and $\mathbf{b}_y$ are parameters for generating word probability distribution, $y_{i,t}$ is index of the maximum probable word, $\mathbf{W}_e$ is a learnable word embedding matrix initialized uniformly, and $\mathbb{O}$ denotes one hot vector.

**Reward Module.** We use automatic metrics CIDEr for computing rewards since recent work on image captioning [28] has shown that CIDEr performs better than many traditional automatic metrics such as BLEU, METEOR and ROUGE. We consider two kinds of reward functions: sentence-level reward and word-level reward. For the $i$-th generated sentence $\mathbf{y}_i = (y_{i,1}, y_{i,2}, \ldots, y_{i,N})$ either from retrieval or generation outputs, we compute a delta CIDEr score at sentence level, which is $R_{sent}(\mathbf{y}_i) = f(\{\mathbf{y}_k\}^i_{k=1}, \text{gt}) - f(\{\mathbf{y}_k\}^{i-1}_{k=1}, \text{gt})$, where $f$ denotes CIDEr evaluation, and gt denotes ground truth report. This assesses the advantages the generated sentence brings in to the existing sentences when evaluating the quality of the whole report. For a single word input, we use reward as delta CIDEr score which is $R_{word}(y_t) = f(\{y_k\}^t_{k=1}, \text{gt}^s) - f(\{y_k\}^{t-1}_{k=1}, \text{gt}^s)$ where $\text{gt}^s$ denotes the ground truth sentence. The sentence-level and word-level rewards are used for computing discounted reward for retrieval policy module and generation module respectively.

## 3.2  Hierarchical Reinforcement Learning

Our objective is to maximize the reward of generated report $\mathbf{Y}$ compared to ground truth report $\mathbf{Y}^*$. Omitting the condition on image features for simplicity, the loss function can be written as:

$$\mathcal{L}(\theta) = -\mathbb{E}_{z,m,y}[R(\mathbf{Y}, \mathbf{Y}^*)] \tag{12}$$
$$\nabla_\theta\mathcal{L}(\theta) = -\mathbb{E}_{z,m,y}[\nabla_\theta \log p(z, m, y)R(\mathbf{Y}, \mathbf{Y}^*)] \tag{13}$$
$$= -\mathbb{E}_{z,m,y}\left[\sum_{i=1}\mathbb{1}(z_i < \frac{1}{2}|z_{i-1})\Big(\nabla_{\theta_r}\mathcal{L}(\theta_r) + \mathbb{1}(m_i = 0|m_{i-1})\nabla_{\theta_g}\mathcal{L}(\theta_g)\Big)\right], \tag{14}$$

where $\theta$, $\theta_r$, and $\theta_g$ denote parameters of the whole network, *retrieval policy module*, and *generation module* respectively; $\mathbb{1}(\cdot)$ is binary indicator; $z_i$ is the probability of topic stop control in Equation 4; $m_i$ is the action chosen by *retrieval policy module* among automatic generation ($m_i = 0$) and all templates ($m_i \in [1, |\mathbb{T}|]$) in the template database. The loss of HRGR-Agent comes from two parts: *retrieval policy module* $\mathcal{L}(\theta_r)$ and *generation module* $\mathcal{L}(\theta_g)$ as defined below.

**Policy Update for Retrieval Policy Module.** We define the reward for retrieval policy module $R^r$ at sentence level. The generated sentence or retrieved template sentence is used for computing the

reward. The discounted sentence-level reward and its corresponding policy update according to REINFORCE algorithm [30] can be written as:

$$R^r(\mathbf{y}_i) = \sum_{j=0}^{\infty} \gamma^j R_{sent}(\mathbf{y}_{i+j}) \tag{15}$$

$$\mathcal{L}(\theta_r) = -\mathbb{E}_{m_i}[R^r(m_i, m_i^*)] \tag{16}$$

$$\nabla_{\theta_r}\mathcal{L}(\theta_r) = -\mathbb{E}_{m_i}[\nabla_{\theta_r}\log p(m_i|m_{i-1})R^r(m_i, m_i^*)], \tag{17}$$

where $\gamma$ is a discount factor; $\mathbf{y}_i$ is the $i$-th generated sequence; and $\theta_r$ represents parameters of retrieval policy module which are $W_u$ and $b_u$ in Equation 5 .

**Policy Update for Generation Module.** We define the word-level reward $R^g(y_t)$ for each word generated by generation module as discounted reward of all generated words after the considered word. The discounted reward function and its policy update for generation module can be written as:

$$R^g(y_t) = \sum_{j=0}^{\infty} \gamma^j R_{word}(y_{t+j}) \tag{18}$$

$$\mathcal{L}(\theta_g) = -\mathbb{E}_{y_t}[R^g(\mathbf{y}_t, \mathbf{y}_t^*)] \tag{19}$$

$$\nabla_{\theta_g}\mathcal{L}(\theta_g) = -\mathbb{E}_{y_t}[\sum_{t=1}\nabla_{\theta_g}\log p(y_t|y_{t-1})R^g(y_t, y_t^*)], \tag{20}$$

where $\gamma$ is a discount factor, and $\theta_g$ represents the parameters of generation module such as $W_y$, $b_y$, $W_e$ in Equation 9-11 and parameters of attention functions in Equation 7 and RNNs in Equation 8. Detailed policy update algorithm is provides in supplementary materials.

## 4  Experiments and Analysis

**Datasets.** We conduct experiments on two medical image report datasets. First, Indiana University Chest X-Ray Collection (IU X-Ray) [8] is a public dataset consists of 7,470 frontal and lateral-view chest x-ray images paired with their corresponding diagnostic reports. Each patient has 2 images and a report which includes impression, findings, comparison and indication sections. We preprocess the reports by tokenizing, converting to lower-cases, and filtering tokens of frequency no less than 3 as vocabulary, which results in 1185 unique tokens covering over 99.0% word occurrences in the corpus.

CX-CHR is a proprietary internal dataset of chest X-ray images with Chinese reports collected from a professional medical institution for health checking. The dataset consists of 35,500 patients. Each patient has one or multiple chest x-ray images in different views such as posteroanterior and lateral, and a corresponding Chinese report. We select patients with no more than 2 images and obtained 33,236 patient samples in total which covers over 93% of the dataset. We preprocess the reports through tokenizing by Jieba [1] and filtering tokens of frequency no less than 3 as vocabulary, which results in 1282 unique tokens.

On both datasets, we randomly split the data by patients into training, validation and testing by a ratio of 7:1:2. There is no overlap between patients in different sets. We predict the 'findings' section as it is the most important component of reports. On CX-CHR dataset, we pretrain a DenseNet with public available ChestX-ray8 dataset [36] on classification, and fine-tune it on CX-CHR dataset on 20 common thorax disease labels. As IU X-Ray dataset is relatively small, we do not directly fine-tune the pretrained DenseNet on it, and instead extract visual features from a DenseNet pretrained jointly on ChestX-ray8 dataset [36] and CX-CHR datasets. Please see Supplementary Material for more details.

**Template Database.** We select sentences in the training set whose document frequencies (the number of occurrence of a sentence in training documents) are no less than a threshold as template candidates. We further group candidates that express the same meaning but have a little linguistic variations. For example, "no pleural effusion or pneumothorax" and "there is no pleural effusion or pneumonthorax" are grouped as one template. This results in 97 templates with greater than 500 document frequency for CX-CHR and 28 templates with greater than 100 document frequency for IU X-Ray. Upon retrieval, only the most frequent sentence of a template group will be retrieved for HRGR-Agent or any rule-based models that we compare with. Although this introduces minor but inevitable error in

the generated results, our experiments show that the error is negligible compared to the advantages that a hybrid of retrieval-based and generation-based approaches brings in. Besides, separating templates of the same meaning into different categories diminishes the capability of *retrieval policy module* to predict the most suitable template for a given visual input, as multiple templates share the exact same meaning. Table 1 shows examples of templates for IU X-Ray dataset. More template examples are provided in supplementary materials.

| Template | df(%) |
|---|---|
| No pneumothorax or pleural effusion. No pleural effusion or pneumothorax. There is no pleural effusion or pneumothorax. | 18.36 |
| The lungs are clear Lungs are clear. The lung are clear bilaterally. | 23.60 |
| No evidence of focal consolidation, pneumothorax, or pleural effusion. no focal consolidation, pneumothorax or large pleural effusion. No focal consolidation, pleural effusion, or pneumothorax identified. | 6.55 |
| Cardiomediastin silhouett is within normal limit. The cardiomediastin silhouett is within normal limit. The cardiomediastin silhouett is within normal limit for size and contour. | 5.12 |

Table 1: Examples of template database of IU X-Ray dataset. Each template is constructed by a group of sentences of the same meaning but slightly different linguistic variations. Top 3 most frequent sentences for a template are displayed in the first and third column. The second column shows document frequency (in percentage of training corpus) of each template.

**Evaluation Metrics.** We use three kinds of evaluation metrics: 1) automatic metrics including CIDEr, ROUGE, and BLEU; 2) medical abnormality terminology detection accuracy: we select 10 most frequent medical abnormality terminologies in medical reports and evaluate average precision and average false positive (AFP) of compared models; 3) human evaluation: we randomly select 100 samples from testing set for each method and conduct surveys through Amazon Mechanical Turk. Each survey question gives a ground truth report, and ask candidate to choose among reports generated by different models that matches with the ground truth report the best in terms of language fluency, content selection, and correctness of medical abnormal finding. A default choice is provided in case of no or both reports are preferred. We collect results from 20 participants and compute the average preference percentage for each model excluding default choices.

**Training Details.** We implement our model on PyTorch and train on a GeForce GTX TITAN GPU. We first train all models with cross entropy loss for 30 epochs with an initial learning rate of 5e-4, and then fine-tune the retrieval policy module and generation module of HRGR-Agent via RL with a fixed learning rate 5e-5 for another 30 epochs. We use 512 as dimension of all hidden states and word embeddings, and batch size 16. We set the maximum number of sentences of a report and maximum number of tokens in a sentence as 18 and 44 for CX-CHR and 7 and 15 for IU X-Ray. Besides, as observed from baseline models which overly predict most popular and normal reports for all testing samples and the fact that most medical reports describe normal cases, we add post-processing to increase the length and comprehensiveness of the generated reports for both datasets while maintaining the design of HRGR-Agent to better predict abnormalities. The post-processing we use is that we first select 4 most commonly predicted key words with normal descriptions by other baselines, then for each key word, if the generated report does not describe any abnormality nor normality of these key words, we add the a corresponding sentence of these key words that describe their normal cases respectively. The key words for IU X-Ray are 'heart size and mediastinal contours', 'pleural effusion or pneumothorax', 'consolidation', and 'lungs are clear'. As observed in our experiments, this step maintains the same medical abnormality term detection results, and improves the automatic report generation metrics, especially on BLEU-n metrics.

**Baselines.** On both datasets, we compare with four state-of-the-art image captioning models: CNN-RNN [34], LRCN [9], AdaAtt [23], and Att2in [28]. Visual features for all models are extracted from the last convolutional layer of pretrained densetNets respectively as mentioned in 4, yielding $16 \times 16 \times 256$ feature maps for both datasets. We use greedy search and argmax sampling for HRGR-Agent and the baselines on both datasets. On IU X-Ray dataset, we also compare with CoAtt [16] which uses different visual features extracted from a pretrained ResNet [11]. The authors of CoAtt [16] re-trained their model using our train/test split, and provided evaluation results for

| Dataset | Model | CIDEr | BLEU-1 | BLEU-2 | BLEU-3 | BLEU-4 | ROUGE |
|---|---|---|---|---|---|---|---|
| **CX-CHR** | CNN-RNN [34] | 1.580 | 0.590 | 0.506 | 0.450 | 0.411 | 0.577 |
| | LRCN [9] | 1.588 | 0.593 | 0.508 | 0.452 | 0.413 | 0.577 |
| | AdaAtt [23] | 1.568 | 0.588 | 0.503 | 0.446 | 0.409 | 0.575 |
| | Att2in [28] | 1.566 | 0.587 | 0.503 | 0.446 | 0.408 | 0.576 |
| | Generation | 0.361 | 0.307 | 0.216 | 0.160 | 0.121 | 0.322 |
| | Retrieval | 2.565 | 0.535 | 0.475 | 0.437 | 0.409 | 0.536 |
| | HRG | 2.800 | 0.629 | 0.547 | 0.497 | 0.463 | 0.588 |
| | HRGR-Agent | **2.895** | **0.673** | **0.587** | **0.530** | **0.486** | **0.612** |
| **IU X-Ray** | CNN-RNN [34] | 0.294 | 0.216 | 0.124 | 0.087 | 0.066 | 0.306 |
| | LRCN [9] | 0.284 | 0.223 | 0.128 | 0.089 | 0.067 | 0.305 |
| | AdaAtt [23] | 0.295 | 0.220 | 0.127 | 0.089 | 0.068 | 0.308 |
| | Att2in [28] | 0.297 | 0.224 | 0.129 | 0.089 | 0.068 | 0.308 |
| | CoAtt* [16] | 0.277 | **0.455** | 0.288 | 0.205 | **0.154** | **0.369** |
| | HRGR-Agent | **0.343** | 0.438 | **0.298** | **0.208** | 0.151 | 0.322 |

Table 2: Automatic evaluation results on CX-CHR (upper part) and IU X-Ray Datasets (lower part). BLEU-n denotes BLEU score uses up to n-grams.

| Dataset | CX-CHR | | | IU X-Ray | | |
|---|---|---|---|---|---|---|
| Models | Retrieval | Generation | HRGR-Agent | CNN-RNN [34] | CoAtt [16] | HRGR-Agent |
| Prec. (%) | 14.13 | 27.50 | **29.19** | 0.00 | 5.01 | **12.14** |
| AFP | 0.133 | 0.064 | **0.059** | 0.000 | **0.019** | 0.043 |
| Hit (%) | – | 23.42 | **52.32** | – | 28.00 | **48.00** |

Table 3: Average precision (Prec.) and average false positive (AFP) of medical abnormality terminology detection, and human evaluation (Hit). The higher Prec. and the lower AFP, the better.

automatic report generation metrics using greedy search and sampling temperature 0.5 at test time. We further evaluated their prediction to obtain medical abnormality terminology detection precision and AFP. Due to the relatively large size of CX-CHR, we conduct additional experiments on it to compare HRGR-Agent with its different variants by removing individual components (Retrieval, Generation, RL). We train a hierarchical generative model (*Generation*) without any template retrieval or RL fine-tuning, and our model without RL fine-tuning (HRG). To exam the quality of our pre-defined templates, we separately evaluate the *retrieval policy module* of HRGR-Agent by masking out the generation part and only use the retrieved templates as prediction (*Retrieval*). Note that *Retrieval* uses the same model as HRG-Agent whose training involves automatic generation of sentences, thus the results of which may be higher than a general retrieval-based system (e.g. directly perform classification among a list of majority sentences given image features).

## 4.1 Results and Analyses

**Automatic Evaluation.** Table 2 shows automatic evaluation comparison of state-of-the-art methods and our model variants. Most importantly, HRGR-Agent outperforms all baseline models that have no retrieval mechanism or hierarchical structure on both datasets by great margins, demonstrating its effectiveness and robustness. On IU X-Ray dataset, HRGR-Agent achieves slightly lower BLEU-1,4 and ROUGE score than that of CoAtt [16]. However, CoAtt uses different pre-processing of reports and visual features, jointly predicts 'impression' and 'findings', and uses single-image input while our method focuses on 'findings' and use combined frontal and lateral view of patients. On CX-CHR, HRGR-Agent increases CIDEr score by 0.73 compared to HRG, demonstrating that reinforcement fine-tuning is crucial to performance increase since it directly optimizes the evaluation metric. Besides, *Retrieval* surpasses *Generation* by relatively large margins, showing that retrieval-based method is beneficial to generating structured reports, which leads to boosted performance of HRGR-Agent when combined with neural generation approaches (*generation module*). To better understand HRGR-Agent's performance, each generated report at testing has on average 7.2 and 4.8 sentences for CX-CHR and IU X-Ray dataset, respectively. The percentages of retrieval vs generation are 83.5 vs 16.5 on the CX-CHR data, and 82.0 vs 18.0 on IU X-Ray, respectively.

**Medical Abnormality Terminology Evaluation.** Table 3 shows evaluation results of average precision and average false positive of medical abnormality terminology detection. HGRG-Agent achieves the highest precision, and is only slightly lower AFP than CoAtt, demonstrating that its robustness on detecting rare abnormal findings which are among the most important components of medical reports.

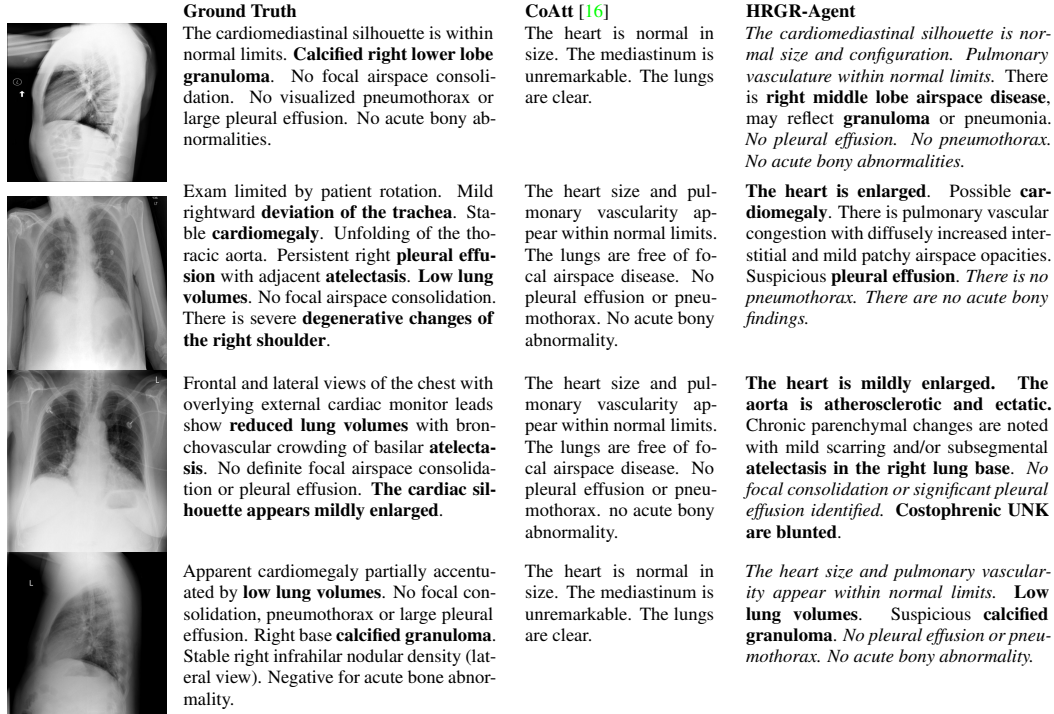

| Ground Truth | CoAtt [16] | HRGR-Agent |
|---|---|---|

The cardiomediastinal silhouette is within normal limits. **Calcified right lower lobe granuloma**. No focal airspace consolidation. No visualized pneumothorax or large pleural effusion. No acute bony abnormalities.

The heart is normal in size. The mediastinum is unremarkable. The lungs are clear.

*The cardiomediastinal silhouette is normal size and configuration. Pulmonary vasculature within normal limits.* There is **right middle lobe airspace disease**, may reflect **granuloma** or pneumonia. *No pleural effusion. No pneumothorax. No acute bony abnormalities.*

---

Exam limited by patient rotation. Mild rightward **deviation of the trachea**. Stable **cardiomegaly**. Unfolding of the thoracic aorta. Persistent right **pleural effusion** with adjacent **atelectasis**. **Low lung volumes**. No focal airspace consolidation. There is severe **degenerative changes of the right shoulder**.

The heart size and pulmonary vascularity appear within normal limits. The lungs are free of focal airspace disease. No pleural effusion or pneumothorax. No acute bony abnormality.

**The heart is enlarged**. Possible **cardiomegaly**. There is pulmonary vascular congestion with diffusely increased interstitial and mild patchy airspace opacities. Suspicious **pleural effusion**. *There is no pneumothorax. There are no acute bony findings.*

---

Frontal and lateral views of the chest with overlying external cardiac monitor leads show **reduced lung volumes** with bronchovascular crowding of basilar **atelectasis**. No definite focal airspace consolidation or pleural effusion. **The cardiac silhouette appears mildly enlarged**.

The heart size and pulmonary vascularity appear within normal limits. The lungs are free of focal airspace disease. No pleural effusion or pneumothorax. no acute bony abnormality.

**The heart is mildly enlarged. The aorta is atherosclerotic and ectatic.** Chronic parenchymal changes are noted with mild scarring and/or subsegmental **atelectasis in the right lung base**. *No focal consolidation or significant pleural effusion identified.* **Costophrenic UNK are blunted.**

---

Apparent cardiomegaly partially accentuated by **low lung volumes**. No focal consolidation, pneumothorax or large pleural effusion. Right base **calcified granuloma**. Stable right infrahilar nodular density (lateral view). Negative for acute bone abnormality.

The heart is normal in size. The mediastinum is unremarkable. The lungs are clear.

*The heart size and pulmonary vascularity appear within normal limits.* **Low lung volumes**. Suspicious **calcified granuloma**. *No pleural effusion or pneumothorax. No acute bony abnormality.*

Figure 3: Examples of ground truth report and generated reports by CoAtt [16] and HRGR-Agent. Highlighted phrases are medical abnormality terms. Italicized text is retrieved from template database.

**Retrieval vs. Generation.** It's worth knowing that on CX-CHR, *Retrieval* achieves higher automatic evaluation scores (Table 2 the $7_{th}$ row) but lower medical term detection precision (Table 3 the $2_{nd}$ column) than *Generation*. Note that *Retrieval* evaluates *retrieval policy module* of HRGR-Agent by masking out the generation results of *generation module*. The result shows that simply composing templates that mostly describe normal medical findings can lead to high automatic evaluation scores since the majority reports describe normal cases. However, this kind retrieval-based approaches lack of the capability of detecting significant but rare abnormal findings. On the other hand, the high medical abnormality term detection precision and low average false positive of HRGR-Agent verifies that its *generation module* learns to describe abnormal findings. The win-win combination of *retrieval policy module* and *generation module* leads to state-of-the-art performance of HRGE-Agent, surpassing a generative model (*Generation*) that is purely trained without any retrieval mechanism.

**Human Evaluation.** Table 3 (last row) shows average human preference percentage of HRGR-Agent compared with *Generation* and CoAtt [16] on CX-CHR and IU X-Ray respectively, evaluated in terms of content coverage, specific terminology accuracy and language fluency. HRGR-Agent achieves much higher human preference than baseline models, showing that it is able to generate natural and plausible reports that are human preferable.

**Qualitative Analysis.** Figure 3 demonstrate qualitative results of HRGR-Agent and baseline models on IU X-Ray dataset. The reports of HRGR-Agent are generally longer than that of the baseline models, and share a well balance of templates and generated sentences. And, among the generated sentences, HRGR-Agent has higher rate of detecting abnormal findings.

# 5 Conclusion

In this paper, we introduce a novel Hybrid Retrieval-Generation Reinforced Agent (HRGR-Agent) to perform robust medical image report generation. Our approach is the first attempt to bridge human prior knowledge and generative neural network via reinforcement learning. Experiments show that HRGR-Agent does not only achieve state-of-the-art performance on two medical image report datasets, but also generates robust reports that has high precision on medical abnormal findings detection and best human preference.

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
