[Supplementary Material]

# Supplementary Material for
# Hybrid Retrieval-Generation Reinforced Agent for Medical Image Report Generation

**Christy Y. Li**[*]
Duke University
yl558@duke.edu

**Xiaodan Liang**[†]
Carnegie Mellon University
xiaodan1@cs.cmu.edu

**Zhiting Hu**
Carnegie Mellon University
zhitingh@cs.cmu.edu

**Eric P. Xing**
Petuum, Inc
epxing@cs.cmu.edu

## 1  Policy Update Algorithm

Here we describe our policy update algorithm for *retrieval policy module* and *generation module*. Note that, if the retrieval policy module predicts a template in $\mathbb{T}$, only retrieval policy module will be updated by sentence-level reward. However, if retrieval policy module predicts automatic generation, both generation module and retrieval policy module are updated by word-level reward and sentence-level reward respectively. We provide the policy update algorithm in Algorithm 1.

## 2  DenseNet Pretraining

We pretrain a DenseNet [1] with publically avaiable ChestX-ray8 dataset [3] on multi-label classification, and fine-tune it on CX-CHR dataset on 20 common thorax disease labels. ChestX-ray8 dataset [3] comprises 108,948 frontal-view X-ray images of 32,717 unique patients with each image labeled with occurrence of 14 common thorax diseases where labels were text-mined from the associated radiological reports using natural language processing. We expand the 14 labels with 6 additional labels text-mined from CX-CHR dataset for fine-tuning. The additional 6 labels are: tortuous aortic sclerosis, bronchitis, calcification, tuberculosis, interstitial lung disease, and patchy consolidation.

We implement our model on PyTorch and train on a single GeForce GTX TITAN GPU. We add an additional lateral layer as in Feature Pyramid Network [2] for the last three dense blocks and additional convolutional layers to transform feature dimension to 256. We extract features from the last convolutional layer of the second dense block which yields $16 \times 16 \times 256$ feature maps. These feature maps contain higher resolution details and more location information without expanding total feature size than features directly extracted from the last layer of DenesNet (e.g., $16 \times 16 \times 1024$ feature maps). We use initial learning rate of 0.1 and multiply by 0.1 every 10 epochs. We train 30 epochs and select the best model by validation performance. The classification model achieves 78.00% AUC score.

## 3  Template Database

Table 1 shows examples of template database of CX-CHR dataset. The template databases are designed by selecting the top most frequent sentences over a threshold in the training corpus and

---

[*]This work was conducted when Christy Y. Li was at Petuum, Inc.
[†]corresponding author

---

**Algorithm 1:** Policy update procedure for HRGR-Agent

---
**Data:** Images $\{I_j\}$
**Result:** Generated report $\mathbf{Y} = (..., \mathbf{y}_i, ...)$

1 CNN extracts visual features;
2 *image encoder* extracts context vector;
3 **for** *time step* $i$ **do**
4     *sentence decoder* generates topic state $\mathbf{q}_i$;
5     *retrieval policy module* generates $m_i$;
6     **if** $m_i == 0$ **then**
7        **for** *time step* $t$ **do**
8           *generation module* generates $y_i$;
9        **end**
10     **else**
11        retrieve template indexed at $m_i$ from template database;
12     **end**
13 **end**
14 **for** *reversed time step* $i$ **do**
15     **if** $m_i == 0$ **then**
16        **for** *reversed time step* $t$ **do**
17           *reward module* computes $R^g(y_i)$;
18           update *generation module* by reward $R^g(y_i)$;
19        **end**
20     **end**
21     *reward module* computes $R^r(\mathbf{y}_i)$;
22     update *retrieval policy module* by reward $R^r(\mathbf{y}_i)$;
23 **end**

| Template | df (%) | Template | df (%) |
|---|---|---|---|
| 双侧肋膈角锐利<br>两侧肋膈角锐利<br>双肋膈角锐利 | 62.50 | 双侧胸廓对称<br>两侧胸廓对称<br>两胸廓对称 | 15.37 |
| 纵隔气管居中<br>气管、纵隔居中<br>气管纵隔居中<br>气管纵膈居中 | 61.30 | 心影大小、形态正常<br>心脏大小、形态正常<br>心脏形态、大小正常<br>心脏外形、大小正常 | 12.87 |
| 双侧膈面光整<br>双侧膈面光滑<br>两侧膈面光滑<br>两膈面光整 | 28.69 | 膈下未见异常密度影 | 31.28 |
| | | 双肺纹理走形自然 | 2.59 |
| | | 两肺纹理增重 | 2.44 |
| | | 所见骨质无明显异常 | 1.83 |

Table 1: Examples of template database of CX-CHR dataset. Each template is constructed by a group of sentences of the same meaning but slightly different expressions. The second and third column display document frequency of individual sentence and template where all its sentences are included respectively. For a selected template at the retrieval step, only the first sentence is returned.
grouping sentences of the same meaning but slightly different language variation. The document frequency threshold for IU X-Ray and CX-CHR dataset is 100 and 500 respectively.