[Reviews · NeurIPS 2018]

Reviewer 1



This paper studies the task of text generation and proposes to use both retrieval and generation for this task. The formulation proposed in the paper jointly trains to trade-off when to retrieve and when to generate. Given this requires making a hard decision between when and what to retrieve vs when to generate, the paper trains this joint network using policy gradients (the classical REINFORCE algorithm). The paper evaluates the proposed method on the task of generating medical image diagnosis reports, and reports improvements in performance over straight up generation or retrieval methods. I like the paper as it feels like a very natural solution to combine retrieval with text generation methods, and the paper proposes a clean formulation for doing this. In my limited knowledge of the field, I am not aware of other works that explicitly do this (and there may be works that I am unaware of). I am also not sure as to how popular is this task in the community, and it may make sense to include results on more standard datasets and tasks (such as Image Captioning).

Reviewer 2



Summary The paper presents an approach using hierarchical reinforcement learning to address the problem of automatically generating medical reports using diagnostics images. The approach first predicts a sequence of hidden states for each sentence, and deicdes when to stop, and a low level model takes the hidden state and either retrieves a sentence and uses it as an output or passes control to a generator which generates a sentence. The overall system is trained with rewards at both sentence level as well as word-level for generation. The proposed approach shows promise over ablations of the proposed model as well as some sensible baseline CNN-RNN based approaches for image captioning. Strengths + Paper provides the experimental details of the setup quite thoroughly. + Paper clearly mentions the hyperparameters used for training. Weaknesses Motivation 1. It would be nice to explain the motivation for the model more clearly. Is the idea that one wants to be able to retrieve the templated sentences and generate the less templated ones? How is mixing generation and retrieval helping with more common sentences vs less common sentences problem? 2. It would be good to perform evaluation with actual patient outcomes instead of evaluation using metrics like CIDEr or even human evaluation on Mechanical Turk. In a lot of ways, it seems important to get validation in terms of patient outcomes instead of M-Turk since that is what matters at the end of the day. It would also be good to take different risk functions (for saying different things) into account in the model. Notation 3. Eqn. 14 and 15, the notation for conditioning inside the indicator variable is somewhat confusing, and there does not seem an obvious need for it to be there. Also, it would be good to make the bracket after I(z_i > 1 / 2) larger to indicate that it is over both the terms in the gradient. Formulation 4. Choice of delta-CIDEr, especially for providing the reward at each timestep of the generation seems very odd and unjustified. Ideally the reward that is optimized for should be the reward provided at the end of an entire sequence CIDEr(w_1, …, w_T, gt), instead of that we are optimizing a proxy, which is the described delta CIDEr score, which might have a problem, that the reward for a word might be low at the given timestep, but might be a part of say, a 4-gram which would get us really high reward (at the sequence level), yet it would not be given importance because the marginal improvement at the world level is low. A more principled approach might be to get intermediate rewards by training a critic using an approach similar to [A]. 5. How long do the reports tend to be? What is the cost for summarizing things incorrectly? It woud be good to do attention on the image and indeed verify that the model is looking at the image when providing the caption. How often does the model retrieve vs generate? (*) *Preliminary Evaluation* The paper is quite thorough in its execution, has a number of sensible baselines and proposes a sensible model for a real-world task. Although the model itself is not very novel, it seems like a decent application paper that is well executed and clear. References: [A]: Bahdanau, Dzmitry, Philemon Brakel, Kelvin Xu, Anirudh Goyal, Ryan Lowe, Joelle Pineau, Aaron Courville, and Yoshua Bengio. 2016. “An Actor-Critic Algorithm for Sequence Prediction.” arXiv [cs.LG]. arXiv. http://arxiv.org/abs/1607.07086. Final Comments -------------------------------------- After having read the other reviews as well as the author rebuttal, I am unchanged in my opinion about this paper. I think it would be a good application paper at NIPS -- the idea of mixing retrieval with generation in the context of the application makes a lot of sense (although the approach itself is not very novel as pointed out by R4 -- copy mechanisms have been used for sequence to sequence models). In light of the thoroughness of the execution, i still lean towards accept, but it would not be too bad to reject the paper either.

Reviewer 3



This paper proposes a model for medical report generation capable of deciding whether to retrieve template sentences or generating new sentences. This model, the HRGR-Agent, achieves the state of the art on two datasets and outperforms existing methods on human evaluations as well. Overall, this paper presents a method that is not particularly novel to a new application domain (medical reports) that contains some interesting challenges (e.g., referencing the location / attributes of the generated findings as mentioned in line 51). Nothing in the method is specific to the medical domain other than the template database, so it's a little strange that it is not evaluated on other tasks, especially since the templates are generated by simple document frequency thresholding (couldn't you also try this for summarization with a larger set of templates?) As such, I'm borderline on the paper's acceptance; it is a well-done application paper but perhaps a little lacking novelty-wise for NIPS. Detailed comments: - The proposed method reminds me of a copy mechanism in principle (a decoder can decide either to produce a token from the vocabulary or copy part of its input). The relationship should be made clear, as the implementation here seems especially similar to that of See et al (ACL 2017). - Why does the generative model fail to detect "effusion" in Fig 1? Was "no effusion" an option for the retrieval system? What if you added common bigrams or phrases (e.g., "mild_effusion") to the vocabulary of the generative model?